# No influence of emotional expression on size underestimation of upright faces

**Eamonn Walsh**[1,2]*, **Jack Whitby**[1], **Yen-Ya Chen**[1], **Matthew R. Longo**[3]

**1** Department of Basic & Clinical Neuroscience, Institute of Psychiatry, Psychology & Neuroscience, King's College London, London, United Kingdom, **2** Cultural and Social Neuroscience Research Group, Institute of Psychiatry, Psychology & Neuroscience, King's College London, London, United Kingdom, **3** Department of Psychological Sciences, Birkbeck, University of London, London, United Kingdom

* eamonn.walsh@kcl.ac.uk

**Data Availability Statement:** Stimuli, raw data, and analysis scripts for the study are available on the Open Science Framework website: https://osf.io/qf96e/.

## Abstract

Faces are a primary means of conveying social information between humans. One important factor modulating the perception of human faces is emotional expression. Face inversion also affects perception, including judgments of emotional expression, possibly through the disruption of configural processing. One intriguing inversion effect is an illusion whereby faces appear to be physically smaller when upright than when inverted. This illusion appears to be highly selective for faces. In this study, we investigated whether the emotional expression of a face (neutral, happy, afraid, and angry) modulates the magnitude of this size illusion. Results showed that for all four expressions, there was a clear bias for inverted stimuli to be judged as larger than upright ones. This demonstrates that there is no influence of emotional expression on the size underestimation of upright faces, a surprising result given that recognition of different emotional expressions is known to be affected unevenly by inversion. Results are discussed considering recent neuroimaging research which used population receptive field (pRF) mapping to investigate the neural mechanisms underlying face perception features and which may provide an explanation for how an upright face appears smaller than an inverted one. Elucidation of this effect would lead to a greater understanding of how humans communicate.

## Introduction

Faces are among the most important cues we use to recognise people and their mental and emotional states. The perception of faces is believed to involve a type of "configural" processing, distinct from the more featural processing used to perceive other types of stimuli [1]. Featural processing generally refers to the basic arrangement of features within a face (e.g., eyes above a nose, above a mouth) [2]. Configural processing generally refers to variations in the spacing between and positioning of the features e.g., the distance between the eyes, and allows for recognition of an individual face and discriminating between different faces [2–4], though see [5] for a counterargument). One signature of this is stimulus inversion, which is thought to disrupt configural processing while leaving featural processing intact [1, 6, 7]. Inversion has a

**Funding:** The author(s) received no specific funding for this work.

**Competing interests:** The authors have declared that no competing interests exist.

variety of effects on face perception, including judgments of identity [3, 8], of emotional expression [9–11], and recognition of facial distortions [12, 13]. Inversion can result in disrupted processing of specific facial features e.g., the perception of the eyes or mouth may be altered when faces are inverted. These disruptions in feature processing could contribute to perceptual differences between upright and inverted faces. According to theoretical accounts of face processing [14, 15], faces are processed holistically, with a focus on the relationships between facial features rather than individual features themselves. Inverted faces disrupt this holistic processing, as the normal upright configuration is distorted. This disruption may lead to a change in how a face is perceived.

One intriguing example of an inversion effect is an illusion reported by Araragi, Aotani, and Kitaoka [16] in which faces appear to be physically smaller when upright than when inverted. We subsequently replicated this effect and showed further that the illusion appears to be highly selective for faces (12). We tested whether similar illusory size effects also held for human bodies and left and right hands—also very familiar and important stimuli in our lives —and everyday inanimate objects, such as cameras and armchairs. We observed no size illusion for hands or everyday objects. However, a reverse size illusion was observed for bodies, so that upright bodies were perceived as larger than their inverted counterparts, i.e., the size illusion was opposite in direction as for faces. When considered together our results indicate that the face, body, and hands produce an illusion, a reverse illusion and no illusion respectively, suggesting that the brain may process each body-part differentially [17]. This illusion and its directionality thus appear to be specific to faces and provides insights into the perceptual and potential neural mechanisms of configural processing.

Visual cortex receptive field (RF) size increases with higher levels of processing [18, 19]. Upright faces activate higher-level visual areas than inverted faces [20, 21], and may involve neuronal populations with larger receptive fields than those involved in processing the same face inverted. In face-selective regions of the ventral visual pathway, population receptive field (pRF) are larger in response to upright than inverted faces [22]. The encoding of upright faces via larger pRFs may activate configural processing allowing higher spatial resolution and face-wide integration of features. Receptive field size dynamics, both at the single unit and population levels in response to face inversion provide a potential neural explanation of the illusion of an upright face appearing smaller than an inverted one. By the same logic, the observed reverse illusion for bodies would be explained by a reversal of this 'RF size and stimulus orientation' relationship due to respective activations of underlying neural populations, i.e. *smaller* RF size for upright bodies and *larger* RF size for inverted bodies.

One important factor modulating the perception of human faces is emotional expression. Multiple lines of evidence have linked the processing of emotion in faces to configural processing. For example, research using the so-called 'composite face illusion' has shown that recognition of emotional expressions is disrupted when the top and bottom halves of a face are misaligned [23]. This misalignment leaves intact the information available in individual facial features, but disrupts the overall spatial configuration. Reduced emotion recognition with misaligned faces thus implicates configural processing. At the same time, the presence of emotional cues in faces increases the magnitude of the composite face illusion, even when irrelevant to the task [24–26]. Similarly, scrambling a face image (which disrupts configural information, while leaving featural information largely intact) had much strong effects on emotion recognition than blurring (which disrupts featural more than configural information) [9]. Emotional facial expressions conveying strong emotions such as anger or fear, capture our attention more effectively than neutral or happy expressions [27–29]. Fearful and angry emotional expressions may better capture attentional processes, leading to increased cognitive processing of facial features, thereby affecting size perception. Object size, along with retinal size,

can modulate attention [30], even during an illusion [31], while attention can also alter length perception [32]. Different emotional expressions involve specific facial configurations and muscle movements that can affect the perceived size of the face e.g., an angry expression can involve lowering of the eyebrows and a reduction of the space between them, while a happy expression can alter mouth shape and cause the cheeks to rise. People move their faces to express emotions [33] and such changes in the shape of facial features and the distances between them might lead to a perceived change in size of a face.

According to holistic accounts, structural relations established between facial features (e.g., the shape of the mouth) help recognition of emotional facial expressions [34–36]. A general finding is that recognition of different emotional expressions is affected unevenly by inversion so that an inversion effect has been observed for anger, disgust and sadness, but less so for happiness [34, 35, 37], indicating an important role for featural processing of facial expressions. A final line of evidence comes from studies showing that the processes involved in emotional expression are affected by inversion. Inversion has been shown in several studies to disrupt the recognition of emotion in faces [9–11, 34]. Notably, inversion has also been found to impair recognition of emotion from body postures, whether shown as still images [38, 39], movies [40, 41], and point-light displays [41–43]. We therefore suspected that the emotional expression displayed by a face would modulate the effect of inversion on perceived face size.

There is also evidence that specific emotions may be processed differently than others. One set of studies has reported an attentional advantage for angry faces, both in behavioural paradigms [44–46] and using event-related potentials (ERPs) [47]. Other studies, however, have reported an advantage for happy faces [48–52]. The reasons for the differences between these studies are unclear. It is possible that different task demands in each experiment may result in heightening sensitivity to negative versus positive emotions. A recent meta-analysis of ERP studies investigating the anger superiority effect found no overall evidence for a difference between angry and happy faces in attentional components such as the N2pc [53]. The N2pc is the N200 posterior-contralateral component typically elicited at visual cortex electrodes between 180 and 300 ms after stimulus onset and emerges in the hemisphere contralateral to the side of an attended stimulus [54, 55]. Notably, however, this analysis did provide evidence that the N2pc is heightened in response to either happy or angry faces compared to neutral faces.

Neuroimaging studies have also found evidence that face-selective regions of the ventral visual pathway such as the fusiform face area are involved in processing emotional expressions. These areas respond more strongly when viewers judge emotional expression [56], show release from adaptation when expression changes [57], and contain dissociable representations of different emotions allowing decoding of expression from fMRI signals [58]. Moreover, there is evidence that visual perception of certain emotions from facial displays may involve specific brain regions. For example, the recognition of facial fear has been linked to the amygdala [59–62] and dorsolateral frontal cortex [63], and fearful faces have been found in several studies to produce larger responses in face-selective visual regions than neutral faces [61, 63–65]. Similarly, facial anger has been linked to the basal ganglia [66], the cingulate gyrus [63], and orbitofrontal cortex [67]. Other emotions have been linked to different brain areas. Sadness has been linked to the amygdala [67, 68] and right temporal pole [67]. In contrast, disgust has been linked to the insula [69, 70].

In this study, we investigated whether the emotional expression of a face modulates the magnitude of the size illusion for upright versus inverted faces. We used a paradigm very similar to that in our previous study [17], while manipulating whether the displayed faces had neutral, happy, afraid, or angry expressions. On each trial, participants saw two faces, one upright and the other inverted, on either side of a fixation cross. The two faces were identical except

for their orientation and size. Across trials, we manipulated the size of the inverted face according to the method of constant stimuli to identify the face that was perceived as equal in size to the upright standard. In line with findings showing that the identification of facial emotional expressions is affected unevenly by inversion, we predicted that the inversion effect which has been reported previously for neutral faces [16, 17] would be larger in magnitude for faces showing emotional expressions, particularly negative emotions (i.e., fear and anger).

## Method

### Participants

Forty individuals in the United Kingdom participated after giving informed consent. Participants were recruited from the Prolific service in 2021. Data from one participant were excluded due to low model fit (see below). The remaining 39 participants (23 women, 15 men, 1 self-described as agender) ranged from 19 to 74 years of age (*M*: 37.5 years, *SD*: 16.0 years). No personally-identifying information was collected about participants. Procedures were approved by the Department of Psychological Sciences Research Ethics Committee at Birkbeck and written consent obtained.

To determine our sample size, we conducted a power analysis using G*Power 3.1 [71] assuming a medium effect size ($d_z = 0.5$) for the paired t-tests comparing the neutral expression with each of the emotional expressions, with power of .80 and alpha of .05. This analysis indicated that 35 participants were needed. We aimed to test 40 participants to provide a buffer in case data needed to be excluded. Because we also ran an analysis of variance (ANOVA) comparing the four conditions, we also conducted a sensitivity analysis to determine the smallest effect size we would have power to detect. This indicated that our final sample of 39 participants had greater than .80 power to detect an effect size of $\eta_p^2 = .04$, a small-to-medium effect by conventional criteria.

### Stimuli

Fig 1 shows examples of the stimuli used in this experiment. Face stimuli were taken from the Karolinska Director Emotional Faces [72]. The neutral, happy, angry, and afraid facial expressions for four men and four women were used. Fig 2 shows four typical experimental trials.

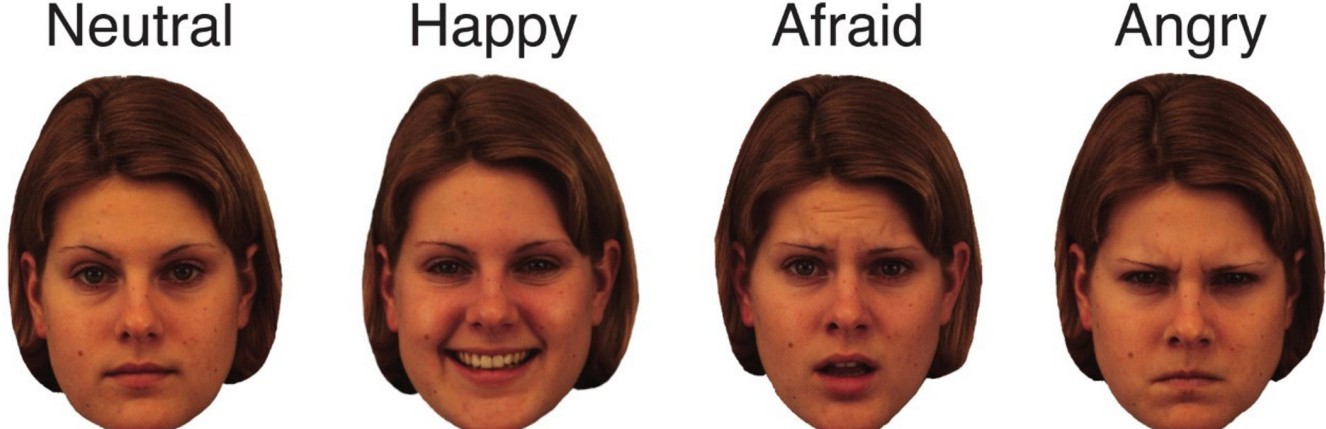

**Fig 1. Example stimuli used in the experiment.** Four emotional expressions (neural, happy, afraid, angry) were shown from four male and four female models.

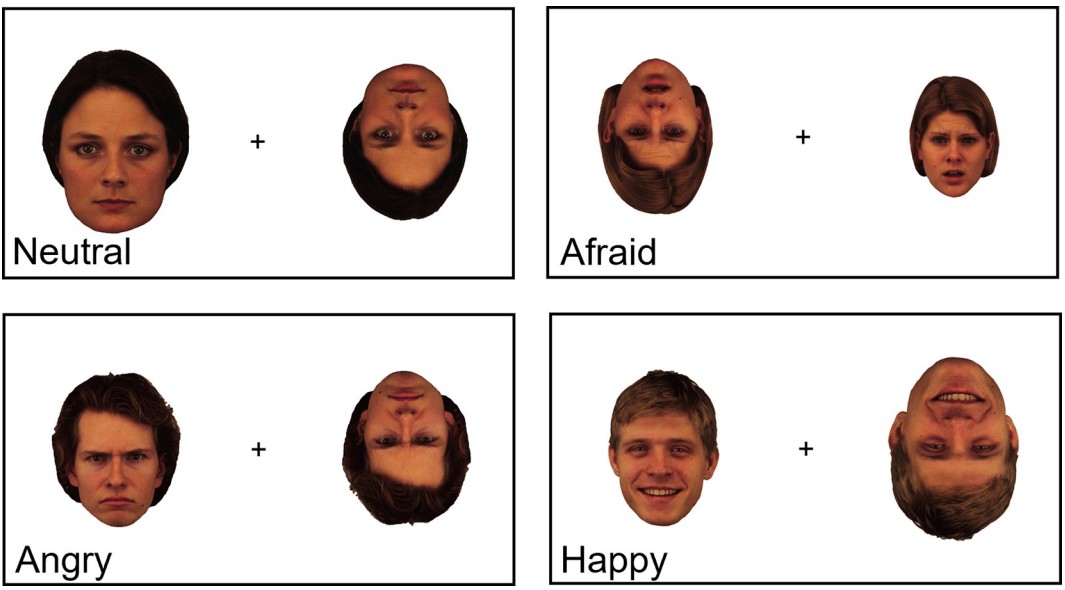

**Fig 2. Schematic showing 4 typical experimental trials from each of the 4 experimental conditions (neutral, afraid, angry, and happy emotional expressions).** Each trial consisted of the same face with the same emotional expression presented on either side of a fixation cross. One image was always inverted, while the other was always upright. One image was always a standard size, while the size of the counterpart image could vary.

## Procedures

The experiment was conducted online using the Gorilla platform [73]. Participants could complete the experiment using desktop, laptop, or tablet computers, but not on mobile phones. Other aspects of the study were closely modelled on our previous study using this paradigm [17].

The upright stimulus was always shown at 500 pixels in height (the physical size and viewing distance depended on the specific computer used by the participant). The size of the inverted stimulus was manipulated across trials according to the method of constant stimuli. Seven sizes of the inverted stimulus were used, corresponding to an increase of the linear dimensions of the image by -9, -6, -3, 0, 3, 6, or 9% (i.e., 455, 470, 485, 500, 515, 530, 545 pixels). This differs from our previous study [17], in which the size of both the upright and inverted stimuli varied across trials. This change was made to simplify the experimental design.

Each trial started with a 500 ms fixation cross, which was followed by the appearance of the faces on either side of the cross. On each trial, the upright and inverted face identity was the same (though the size could differ), and with the same emotional expression. The stimuli remained on the screen until the participant responded. The participant's task was to indicate which stimulus appeared physically larger, by clicking the mouse cursor (if using a desktop or laptop) or tapping it with their finger (if using a tablet).

There were eight blocks of trials, two of each of the four emotional expressions. The first four blocks consisted of one of each facial expression, counterbalanced across participants according to a Latin square. The final four blocks were in the reverse order from the first four. Each block consistent of 56 trials, formed by 8 repetitions of each of the 7 inverted stimulus sizes. Of these 8 repetitions, half had the upright stimulus on the right side of the screen, and half on the left. The order of the 56 trials was randomized for each block.

Stimuli, raw data, and analysis scripts for the study are available on the Open Science Framework website: https://osf.io/qf96e/

## Analysis

The analysis was similar to that we used in our previous paper [17]. For each participant we fit a psychometric function to data from each emotional expression separately, using the Palamedes toolbox [74] for MATLAB (Mathworks, Natick, MA). We estimated the parameters of the best-fitting cumulative Gaussian function using maximum-likelihood estimation. For each curve, we obtained two parameters, the point-of-subjective equality (PSE), the mean of the Gaussian, and the slope, or the inverse of the standard deviation of the Gaussian. We also calculated the $R^2$ value of the curve as a measure of goodness-of-fit. We excluded from the analysis any participant for which $R^2$ was less than .5 in any of the four emotional expressions. As mentioned above, one participant was excluded on this basis.

The key parameter for assessing the illusion is the PSE, which quantifies the difference in size between the inverted and upright stimuli for which the stimuli are perceived as being the same size. Positive PSEs indicate a bias to perceive upright stimuli as larger than inverted stimuli, while negative PSEs indicate the opposite.

We used repeated-measures analysis of variance (ANOVA) to compare the four conditions on $R^2$, PSE, and slope. Where Mauchley's test indicated violation of the sphericity assumption, the Greenhouse-Geisser correction was applied. For F-tests, $\eta_p^2$ is used as a measure of effect size. For key tests, we also used Bayesian ANOVA [75] in JASP 0.16.1 to calculate Bayes Factors in order to quantify evidence for or against the null hypothesis. The presence of an illusion was confirmed in each condition individually, using one-sample t-tests to compare the mean PSE to 0. For t-tests, Cohen's d is used as a measure of effect size.

## Results

Results are shown in Fig 3. There was good overall fit of the psychometric functions to the data, with mean $R^2$ values of .968, .950, .955, and .964 in the neutral, happy, afraid, and angry conditions, respectively. The $R^2$ values did not differ significantly across conditions, $F(2.38, 90.32) = 2.05$, $p = .127$, $\eta_p^2 = .051$.

To assess the basic size illusion, we compared the mean PSE in each condition to 0. Clear biases to perceive the upright face as smaller than the inverted face were found in all four conditions: neutral faces ($M$: -1.21%), $t(38) = -3.80$, $p < .001$, $d = 0.609$; happy faces ($M$: -1.35%), $t(38) = -3.81$, $p < .001$, $d = 0.610$; afraid faces ($M$: -1.26%), $t(38) = -3.11$, $p < .005$, $d = 0.498$; and angry faces ($M$: -1.68%), $t(38) = -4.91$, $p < .0001$, $d = 0.787$. This provides a clear replication of the illusion reported in previous studies [16, 17, 76].

We next compared PSEs for each of the emotional expressions to the neutral expression using planned paired t-tests. No significant differences from the neutral expression were found for happy, $t(38) = 0.49$, $p = .63$, $d_z = 0.079$; afraid, $t(38) = 0.17$, $p = .87$, $d_z = 0.027$; or angry, $t(38) = 1.93$, $p = 0.061$, $d_z = 0.309$; expressions.

An ANOVA on PSE values provided no evidence for a difference in the magnitude of the illusion as a function of emotion expression, $F(3, 114) = 0.94$, $p = .422$, $\eta_p^2 = .024$. A Bayesian ANOVA provided substantial support for the null hypothesis of no difference across conditions, $BF_{01} = 9.913$.

An ANOVA on slope values similarly provided no evidence for any difference between emotional expressions, $F(3, 114) = 0.72$, $p = .543$, $\eta_p^2 = .019$. A Bayesian ANOVA again provided strong support for the null hypothesis of no difference between conditions, $BF_{01} = 12.947$.

The magnitude of the illusion was correlated across participants for all pairs of the four conditions, with Pearson's correlation coefficients ranging from .495 to .730, all of which were significant (all $p$'s < .002), using Holm-Bonferroni correction for multiple comparisons.

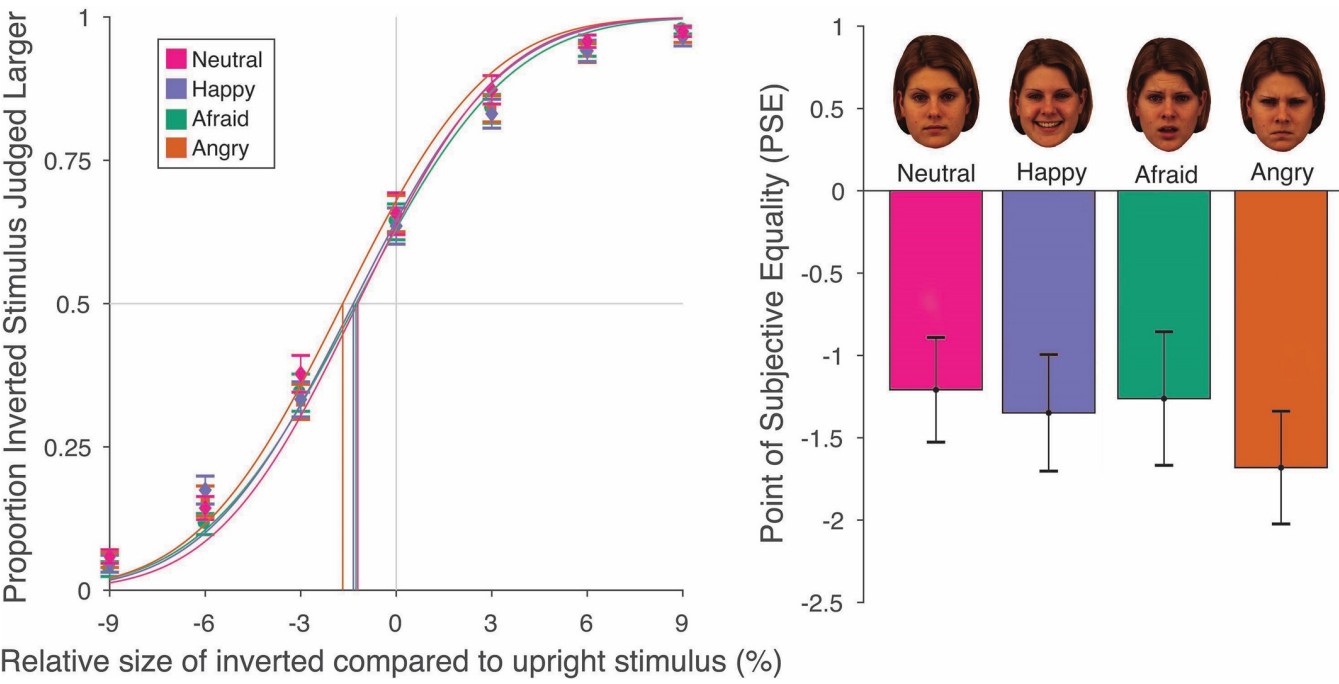

**Fig 3.** *Left panel*: The proportion of trials for which the inverted face was judged as larger than the upright face, as a function of the size of the inverted stimulus in comparison to the upright one. *Right panel*: Mean PSE values for each emotional expression. For all four expressions, there was a clear bias for inverted stimuli to be judged as larger than upright ones, thereby demonstrating the size underestimation of upright faces. Error bars are one standard error.

The age range in the present study is comparatively large relative to studies in experimental psychology, which often include mainly 18–22 year-old university students. We thus investigated the relation between the size inversion illusion and age in our sample. We averaged PSE values across the 4 conditions to calculate a single measure of illusion magnitude for each participant. This showed a modest positive correlation with age, $r(37) = .339$, $p < .05$. This indicates that the magnitude of the illusion decreases with age.

## Discussion

The present results provide a clear replication of the illusion for upright faces to be perceived as physically smaller than inverted faces, which has been reported in recent studies [16, 17, 76]. This effect was clearly apparent in all four conditions in the present experiment. There was no evidence, however, that the magnitude of this illusion was affected by the emotional expression displayed by the face.

In previous research, emotional expressions have been found to modulate several aspects of face perception, including holistic processing in the composite face illusion [24–26], attentional prioritization [45, 48, 52], and activation of face-selective regions of the ventral visual pathway [56, 58]. Such effects, however, have often been inconsistent across studies. For example, while some studies have reported that angry faces have privileged access to attention [44–46], other studies have instead reported that it is happy faces that have privileged access [50–52]. In this light, the null findings in the present study should perhaps not be deeply surprising.

While the Karolinska faces which we used in this study have been widely used, and extensively validated, for research on visual perception of facial emotional expressions, they do lack dynamic cues that are present in our real interactions with others. There is substantial evidence

that videos of emotional expressions, which including more dynamic information unfolding over time, are processed differently than still images [77]. It is possible that effects of emotional expression might emerge if richer, more dynamic stimuli were employed.

Recent neuroimaging research has used population receptive field (pRF) mapping to investigate the neural mechanisms underlying spatial integration of face features that may comprise configural processing. Poltoratski and colleagues [22] found that in face-selective regions of the ventral visual pathway, pRFs were larger in responses to faces presented upright than inverted. This finding is intuitively unexpected. In general, small RFs are associated with regions of high spatial sensitivity [78]. The same is true for pRFs, which are much smaller in regions representing the fovea than the visual periphery [79, 80]. Given the salience of faces, and the extent to which face perception relies on dedicated brain areas, we might have expected small pRFs for faces. Poltoratski and colleagues interpret the larger pRFs for upright faces as a mechanism underlying configural processing. Whereas featural processing relies on high spatial acuity, configural processing in contrast relies on high spatial integration from across the face. This face-wide integration is served by an increase in pRF size. Intriguingly, however, there is an inverse relation between the size of pRFs and perceived object size [81], with objects appearing larger on regions with small pRFs (e.g., the fovea) than on regions with larger pRFs (e.g., the periphery). The changes in pRF size with face inversion [22] thus provides a potential mechanism underlying the illusion of upright faces appearing smaller than inverted ones [16, 17, 76]. Our results show that such a mechanism is not further modulated by the emotional content of the stimulus and the processing of facial emotional expressions relies on spatial integration across facial features rather than on the processing of individual facial features.

In conclusion, the present results provide a clear replication of the size illusion for upright versus inverted faces which has been reported previously [16, 17]. This effect does not appear to be modulated by the presence of emotional expressions in the face. Nevertheless, it does appear to be highly specific for faces, and does not occur for other categories of object [17]. The illusion thus appears linked to the basic structural encoding of faces and offers a means to a better understanding of how humans communicate.

## Acknowledgments

For the purposes of open access, the author has applied a Creative Commons Attribution (CC BY) licence to any Accepted Author Manuscript version arising from this submission.

## Author Contributions

**Conceptualization:** Matthew R. Longo.

**Data curation:** Matthew R. Longo.

**Formal analysis:** Matthew R. Longo.

**Methodology:** Eamonn Walsh.

**Project administration:** Jack Whitby, Yen-Ya Chen.

**Software:** Eamonn Walsh.

**Supervision:** Eamonn Walsh, Matthew R. Longo.

**Writing – original draft:** Matthew R. Longo.

**Writing – review & editing:** Eamonn Walsh, Jack Whitby, Yen-Ya Chen, Matthew R. Longo.

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
