## [Decision Letter · Decision Letter 0]

4 Jul 2023

PONE-D-23-10095No Influence of Emotional Expression on Size Underestimation of Upright FacesPLOS ONE

Dear Dr. Walsh,

Thank you for submitting your manuscript to PLOS ONE. After careful consideration, we feel that it has merit but does not fully meet PLOS ONE’s publication criteria as it currently stands. Therefore, we invite you to submit a revised version of the manuscript that addresses the points raised during the review process. Your manuscript has been reviewed by two experts in the field. While reviewers consider your study interesting, they have also raised a few critical concerns. In particular, one reviewer has raised concerns over the methodology and suggested control experiments and analysis to further strengthen the conclusion of the negative results. We want to mention that, while many scientists believe that a null result should be considered unpublishable (not being defined as a scientific discovery), PLOS ONE is open to negative findings to avoid publication bias. Yet, we should always pay extra attention to the experimentation and interpretation of these null findings. We, therefore, recommend the authors to adequately address the reviewers' concerns.

We look forward to receiving your revised manuscript.

Kind regards,

Shenbing Kuang, Ph.D.

Academic Editor

PLOS ONE

3. We note that Figures 1 and 2 in your submission contain copyrighted images. All PLOS content is published under the Creative Commons Attribution License (CC BY 4.0), which means that the manuscript, images, and Supporting Information files will be freely available online, and any third party is permitted to access, download, copy, distribute, and use these materials in any way, even commercially, with proper attribution. For more information, see our copyright guidelines: http://journals.plos.org/plosone/s/licenses-and-copyright.

a. You may seek permission from the original copyright holder of Figures 1 and 2  to publish the content specifically under the CC BY 4.0 license.

Reviewers' comments:

Reviewer's Responses to Questions

**Comments to the Author**

1. Is the manuscript technically sound, and do the data support the conclusions?

Reviewer #1: Partly

Reviewer #2: Partly

2. Has the statistical analysis been performed appropriately and rigorously? 

Reviewer #1: Yes

Reviewer #2: Yes

3. Have the authors made all data underlying the findings in their manuscript fully available?

Reviewer #1: Yes

Reviewer #2: Yes

4. Is the manuscript presented in an intelligible fashion and written in standard English?

Reviewer #1: Yes

Reviewer #2: No

5. Review Comments to the Author

Reviewer #1: The authors report a study conducted online aimed at assessing whether the magnitude of the face size illusion -- upright faces are judged smaller than inverted faces -- is modulated by the expression shown by faces. A pair of faces – one upright of constant size and the other inverted of size varying between -9 to + 9 in steps of 3 – were presented left and right of fixation and remained on screen until response.

Participants (N= 40) indicated which face was larger. Participants completed 8 blocks of 56 trials (8 repetition of each of the 7 inverted face sizes), 2 blocks for each of the 4 facial expressions (neutral, happy, angry, afraid). Findings showed that the face size illusion was similar for all faces, regardless of the facial expression. The Ms is well written and the research question is well described. There are a few aspects that require additional information. More specifically:

1) It would help the reader if there was a figure with examples of the inverted faces in different size and the display with the pair of faces.

2) The authors report the power analysis conducted with G*Power to determine the sample size for a t-test (page 6) however results of ANOVA for repeated measures are reported (page 8). Should the authors rather report a power sensitivity analysis? That is what is the smallest difference that could be detected with 39 participants?

3) Participants’ age ranged from 19 to 74 years, which is rather unusual considering that there are age-related changes in processing faces and emotional expressions from faces. Depending on how many participants are aged over 55/60, it would be helpful also to report the PSEs for young and old participants.

4) In the discussion, the potential neural mechanism underlying the face size illusion. It would be very helpful if this was mentioned in the introduction.

5) Is the face size illusion the same for inverted faces presented on the left and for inverted faces presented on the right? It would be interesting if the PSEs could be presented also as a function of where the inverted face was presented (left/right) and if this aspect could be acknowledged in the discussion.

Reviewer #2: General:

This study utilised a paradigm based on the Method of Constant Stimuli, investigated the illusion of underestimating sizes of upright faces versus their inverted counterparts, and further compared the magnitudes of this illusion across four facial expressions including neutral, happy, fearful, and angry. They found that the size underestimation for upright faces existed across all four emotions, however, the magnitude of illusions did not differ. Their main conclusion therefore is that the emotional expressions have no influence on size underestimation of upright faces.

Although this topic itself is interesting, and I personally think it is valuable to lead to further investigations even on the neural level. The current paper did not provide strong justification for this value, nor in-depth discussion of the plausible reasons for their findings. I found the writing sometimes oversimplifies the relevant concepts, especially in the Introduction, where lots of terms were used as if they are common knowledge when they are not, such as “attentional component N2pc”. Major effort would be required for improving the Introduction. I also have a major concern (2nd point below) regarding the analysis/methodology, although I’m fully open to change my mind, I would like to see strong argument for it. Overall, I cannot accept the paper for how it is now, but I can change opinion if my concerns are addressed well.

Major:

- Throughout the introduction, it didn’t connect clearly regarding why emotional expressions and the inversion size illusion should be relevant, and why it is important to investigate this question. There was only one sentence linking the two concepts – “ we therefore suspect that the emotional expression displayed by a face….on perceived face size.” However, I would be also interested in information particularly for this size illusion by inversion. Such as the practical relevance of the illusion/why it is important or interesting to investigate; also, why we must know if emotional expressions would modulate the illusion or not (i.e., any practical meanings? or how can it help advancing the literature?). The motivation of the current research question was not explained well, as it read like the introduction only explained why emotional expressions are special in human faces perception. More depth is needed, and the logical flow should be revised and strengthened.

- Based on my understanding, the DVs in the analysis are all about the illusion between an upright and an inverted face, and the magnitudes of illusion were compared across emotions, in order to see the effects of emotional expression on the illusion. If this understanding was correct, then I have a major problem with the overall methodology, or at least the delivery of justification.

o Inversion commonly impairs recognition of a face and emotional expressions, especially at the early perceptual stages. Therefore, manipulation of emotional expressions should only be valid in upright faces. Across the inverted faces, even though the presentations of each emotion might be different to certain extent, it should be mainly due to low-level visual features instead of the emotion per se. In

other words, theoretically, there was cognitively no difference of emotional expressions across inverted faces.

o As such, it seems that if the emotional expressions could modulate perceived size of a face, the face would be in upright orientation. This consequently affects the magnitudes of the illusion associated with inversion.

o If the above line of thoughts is agreed by the authors, then what seems insufficient should be in the analysis – perceived sizes across emotions should be compared within the upright faces, and inverted faces respectively (in this case, also varying the size of upright faces seem to provide a more balanced comparison). Then the conclusion of “no influence of emotional expression on size underestimation of upright faces” can be drawn with more confidence, alone with more in-depth justification/explanation.

Minor:

Introduction:

- Page 3, line 2 – brief explanations for “configural processing” and “featural processing” would make the following points easier to follow.

- Page 3 – “inversion has a variety of effects on face perception…”, better to specify the sort of effects, e.g. disruption.

- Page 3, end of 1st paragraph – did you find reverse illusion for body as well, then why the following sentence claimed this effect tend to be specific for faces? There needs a bit more clarification.

- Page 4, 2nd paragraph – information about the N2pc seems too little, and corresponding interpretation of results from the cited paper was insufficient. Hard to see the necessity of including this reference.

Method:

- For the power analysis, it would be better to specify what are “the t-tests” (e.g. one-way, within-subject, etc”. I just wasn’t sure about what happened here based on the text.

- For each trial, the upright and inverted faces were both in the same expressions, right? Please specify this.

Overall:

- The paper mentioned “as in our previous study” multiple times, there should be more detailed information regarding what is in the previous paper to help with reading of the current one. If the current paper already provided enough information, then maybe just reduce mentions of the previous paper.

6. PLOS authors have the option to publish the peer review history of their article (what does this mean?). If published, this will include your full peer review and any attached files.

Reviewer #1: No

Reviewer #2: No

---

## [Author Response · Author response to Decision Letter 0]

10 Sep 2023

Reviewer #1: The authors report a study conducted online aimed at assessing whether the magnitude of the face size illusion -- upright faces are judged smaller than inverted faces -- is modulated by the expression shown by faces. A pair of faces – one upright of constant size and the other inverted of size varying between -9 to + 9 in steps of 3 – were presented left and right of fixation and remained on screen until response. 

Participants (N= 40) indicated which face was larger. Participants completed 8 blocks of 56 trials (8 repetition of each of the 7 inverted face sizes), 2 blocks for each of the 4 facial expressions (neutral, happy, angry, afraid). Findings showed that the face size illusion was similar for all faces, regardless of the facial expression. The Ms is well written and the research question is well described. There are a few aspects that require additional information. More specifically: 

1) It would help the reader if there was a figure with examples of the inverted faces in different size and the display with the pair of faces. 

Authors’ response: 

Thank you for this suggestion. We have now created a new figure (Figure 2) and caption showing example trials with pairs of faces in different sizes and from each emotion condition (neutral, afraid, angry, and happy). We have renamed the erstwhile Figure 2 (Results section) as Figure 3.

2) The authors report the power analysis conducted with G*Power to determine the sample size for a t-test (page 6) however results of ANOVA for repeated measures are reported (page 8). Should the authors rather report a power sensitivity analysis? That is what is the smallest difference that could be detected with 39 participants? 

Authors’ response:

This is a fair point. We chose to use the more focused pairwise t-tests for our power analysis because the effect in ANOVA is quite non-specific. Nevertheless, we acknowledge that it is strange that the pairwise t-tests that we based the power analysis on were not actually reported in the paper, given that the omnibus effect in the ANOVA was not significant. We do not feel that it is appropriate to completely replace our previous power analysis, as it was in fact what we used to determine our sample size. We have made the following changes to the manuscript in response to this point:

1. We now report the paired t-tests comparing PSEs for each of the emotional expressions (happy, afraid, angry) to neutral (pg. 8). We feel that this is appropriate even though the omnibus ANOVA is not significant given that these were planned comparisons on which we based the power analysis to determine our sample size. None of these t-tests is significant.

2. We now report a sensitivity power analysis on the ANOVA, in addition to our a priori power analysis on the paired t-tests which we reported in the previous version of the paper. The new paragraph reporting these analyses reads as follows (pg. 8):

“To determine our sample size, we conducted a power analysis using G*Power 3.1 ([Faul et al.]) assuming a medium effect size (dz = 0.5) for the paired t-tests comparing the neutral expression with each of the emotional expressions, with power of .80 and alpha of .05. This analysis indicated that 35 participants were needed. We aimed to test 40 participants to provide a buffer in case data needed to be excluded. Because we also ran an analysis of variance (ANOVA) comparing the four conditions, we also conducted a sensitivity analysis to determine the smallest effect size we would have power to detect. This indicated that our final sample of 39 participants had greater than .80 power to detect an effect size of ηp2 = .04, a small-to-medium effect by conventional criteria.”

3) Participants’ age ranged from 19 to 74 years, which is rather unusual considering that there are age-related changes in processing faces and emotional expressions from faces. Depending on how many participants are aged over 55/60, it would be helpful also to report the PSEs for young and old participants. 

Authors’ response:

We did not apply any exclusion criteria except that participants had to be 18 years of age or older. We understand the reviewer’s point, but respectfully we feel that the age range of our participants is large only relative to a default assumption that participants in psychology experiments are 18-22 year-old University students. In our view, it is that default that is problematic, not our use of a more diverse, generalisable age range.

We have now calculated the correlation between age and PSEs (collapsed across the four conditions). We find a modest positive correlation, r(37) = .339, p < .05, indicating that the magnitude of the inversion illusion decreases with age. We now report this in the results section (pg. 12 - 13):

“The age range in the present study is comparatively large relative to studies in experimental psychology, which often include mainly 18-22 year-old university students. We thus investigated the relation between the size inversion illusion and age in our sample. We averaged PSE values across the 4 conditions to calculate a single measure of illusion magnitude for each participant. This showed a modest positive correlation with age, r(37) = .339, p < .05. This indicates that the magnitude of the illusion decreases with age.”

4) In the discussion, the potential neural mechanism underlying the face size illusion. It would be very helpful if this was mentioned in the introduction. 

Authors’ response:

Thank you for this suggestion. We now introduce the potential neural mechanism underlying the face size illusion in a new section on Page 4 of the introduction.

5) Is the face size illusion the same for inverted faces presented on the left and for inverted faces presented on the right? It would be interesting if the PSEs could be presented also as a function of where the inverted face was presented (left/right) and if this aspect could be acknowledged in the discussion. 

Authors’ response:

Respectfully, we do not see any rationale for this analysis. The Reviewer does not provide any theoretical basis for comparing the left versus right hemifields. Moreover, this variable is completely counterbalanced in our experimental design, and so is not a confound in any of our analyses. We thus do not feel that this needs to be discussed here. 

We can imagine that there could conceivably be hemispheric differences in visual size perception, and that the present dataset could provide insight into this. We have made our raw data freely available on the Open Science Framework website (pg. 10 of manuscript), so researchers interested in following up this separate question can analyse the data. However, we do not feel that this analysis is appropriate or relevant for the current paper given that, though interesting, it is exploratory and has no direct connection to the specific research questions that motivate our current study.

The authors thank Reviewer 1 for their helpful comments which help strengthen our paper.

 

Reviewer #2: General: 

This study utilised a paradigm based on the Method of Constant Stimuli, investigated the illusion of underestimating sizes of upright faces versus their inverted counterparts, and further compared the magnitudes of this illusion across four facial expressions including neutral, happy, fearful, and angry. They found that the size underestimation for upright faces existed across all four emotions, however, the magnitude of illusions did not differ. Their main conclusion therefore is that the emotional expressions have no influence on size underestimation of upright faces. 

Although this topic itself is interesting, and I personally think it is valuable to lead to further investigations even on the neural level. The current paper did not provide strong justification for this value, nor in-depth discussion of the plausible reasons for their findings. I found the writing sometimes oversimplifies the relevant concepts, especially in the Introduction, where lots of terms were used as if they are common knowledge when they are not, such as “attentional component N2pc”. Major effort would be required for improving the Introduction. I also have a major concern (2nd point below) regarding the analysis/methodology, although I’m fully open to change my mind, I would like to see strong argument for it. Overall, I cannot accept the paper for how it is now, but I can change opinion if my concerns are addressed well. 

Authors’ response:

Thank you for this point. We have added new sections to the Introduction of the manuscript (pages 3-6). We had included a reference for a review paper (Liu et al,, 2020) for the interested reader immediately after our mention of the N2pc, which defines and discusses the N2pc in considerable detail. Further to your comment, we have now also added a definition for the N2pc within the introduction (Page 6) and we have added 2 extra references to further support this.

Review Reference: Liu Y, Wang Y, Gozli DG, Xiang YT, Jackson T. Current status of the anger superiority hypothesis: A meta-analytic review of N2pc studies. Psychophysiology. 2020;58:e13700. 

Major: 

- Throughout the introduction, it didn’t connect clearly regarding why emotional expressions and the inversion size illusion should be relevant, and why it is important to investigate this question. There was only one sentence linking the two concepts – “ we therefore suspect that the emotional expression displayed by a face….on perceived face size.” However, I would be also interested in information particularly for this size illusion by inversion. Such as the practical relevance of the illusion/why it is important or interesting to investigate; also, why we must know if emotional expressions would modulate the illusion or not (i.e., any practical meanings? or how can it help advancing the literature?). The motivation of the current research question was not explained well, as it read like the introduction only explained why emotional expressions are special in human faces perception. More depth is needed, and the logical flow should be revised and strengthened. 

 Authors’ response:

Thank you. The Reviewer raises a fair point, and we now recognise that we should have made the connection between emotional expressions and the inversion size illusion clearer, and also why it is important to investigate this question. Emotional facial expressions conveying strong emotions such as anger or fear, can capture our attention more than neutral or happy expressions, which can lead to altered cognitive processing of faces. It is known that attention interacts with perceptions of size. Also, different emotional expressions involve specific facial configurations, and more than 40 different facial muscles can be activated alone or in combination which might affect the perceived size of the face. An angry expression, for example, can involve lowering of the eyebrows and a reduction of the space between them, while a happy expression can alter mouth shape and cause the cheeks to rise. People move their faces and change the shape of some of their features and the distances between them, when expressing emotions and these changes may also lead to perceptions of size change of a face. Also, it is known that recognition of emotional expression shows an uneven inversion effect, suggesting holistic processing. We have added a new section to Pages 5-6 to explain this. We have also added a phrase to the Abstract and modified a sentence in the final paragraph of the introduction (Pg. 7) to make this clearer. 

- Based on my understanding, the DVs in the analysis are all about the illusion between an upright and an inverted face, and the magnitudes of illusion were compared across emotions, in order to see the effects of emotional expression on the illusion. If this understanding was correct, then I have a major problem with the overall methodology, or at least the delivery of justification. 

o Inversion commonly impairs recognition of a face and emotional expressions, especially at the early perceptual stages. Therefore, manipulation of emotional expressions should only be valid in upright faces. Across the inverted faces, even though the presentations of each emotion might be different to certain extent, it should be mainly due to low-level visual features instead of the emotion per se. In other words, theoretically, there was cognitively no difference of emotional expressions across inverted faces. 

o As such, it seems that if the emotional expressions could modulate perceived size of a face, the face would be in upright orientation. This consequently affects the magnitudes of the illusion associated with inversion. 

o If the above line of thoughts is agreed by the authors, then what seems insufficient should be in the analysis – perceived sizes across emotions should be compared within the upright faces, and inverted faces respectively (in this case, also varying the size of upright faces seem to provide a more balanced comparison). Then the conclusion of “no influence of emotional expression on size underestimation of upright faces” can be drawn with more confidence, alone with more in-depth justification/explanation. 

Authors’ response:

The Reviewer states correctly that the dependent variable is the perceived difference in size (illusion) between an upright and the same face inverted, and that the magnitude of the size illusion was compared across faces to see if emotional expression influences the illusion. Our aim was to ask whether the emotional expression of a face modulates a specific size illusion which has been reported in previous research. Even if, as the Reviewer states, emotional expressions are more easily differentiated when upright than inverted, it still remains that to quantify the illusion each upright face needs to be compared with an inverted stimulus which is physically identical to the upright stimulus in all respects, aside from size.

In relation to the point that inversion impairs recognition of a face and emotional expressions, especially at the early perceptual stages, we point out that face inversion effects are observed at both early and late perceptual stages, involving multiple levels of face processing (e.g., Caharel, et al., 2006; Eimer, 2000; Henson, et al., 2003). At early perceptual stages, the face inversion effect involves processes such as edge detection, contour integration, and basic feature extraction (N170; Hinojosa, et al., 2015; Itier, & Taylor, 2004). Inversion can disrupt these featural processes, making it more challenging to extract and integrate low-level visual information from a face, contributing to a decreased ability to recognise and discriminate inverted faces. The face inversion effect also involves higher-level perceptual stages, especially associated with configural or holistic face processing, which involves the integration of facial features and the extraction of the spatial relationships between them. Inverted a face disrupts configural processing impairing the ability to process and recognise inverted faces holistically, thereby resulting in reduced recognition accuracy and reduced recognition of emotional expression. Neuroimaging studies indicate that both early and late perceptual stages are involved in face processing, from primary visual cortex (V1), the lateral occipital cortex, to fusiform face area, and the superior temporal sulcus. Face recognition takes at least 400 ms (Anaki, & Bentin, 2009; Barragan-Jason, et al., 2013). The face inversion effect involves a cascade of processing stages, not just early visual analysis but also higher-level configural and holistic processing. 

We believe we have now made clear in our revised manuscript, the need to compare upright with inverted faces. The essential comparison is between an upright and inverted faces, and for the illusion to be experienced. As stated, one might expect uneven face size illusions for the different emotional expressions due to differing configurations of facial features and likely differences in attentional processing between the emotional expressions. Inversion affects emotion recognition of human faces, particularly for complex emotions such as fear, as opposed to happiness. Theories explaining these inversion effects argue that when a face is inverted, holistic processing of facial features is disrupted, making it more difficult for the brain to integrate information from different parts of the face. This, in turn, affects processing of emotional expression. When processing inverted faces, the brain may process individual facial features, which are less informative for identifying emotional expressions.

The suggestion to perceive size across pairs of upright and inverted faces respectively is interesting. 

However, presenting two faces in the same orientation raises issues and can involve their own illusions. When two (upright) images are placed side-by-side, illusions of perspective and size can arise (Kingdom, Yoonessi, & Gheorghiu, 2007). When two identical faces are aligned vertically, the bottom face appears fatter than the upper one (Tomonaga, 2015; Galusca, et al., 2022). When two or more objects including faces, are placed side-by-side there is a tendency for the visual system to treat them as part of the same scene and to process relations between two or more faces (Galusca, et al., 2022). Thus, while we agree that it could be interesting to compare the size of upright faces directly to each other, we feel that this is a different question, and beyond the scope of the present paper.

References

Anaki, D., and Bentin, S. (2009). Familiarity effects on categorization levels of faces and objects. Cognition 111, 144–149.

Barragan-Jason, G., Besson, G., Ceccaldi, M., & Barbeau, E. J. (2013). Fast and famous: looking for the fastest speed at which a face can be recognized. Frontiers in psychology, 4, 100.

Caharel, S., Fiori, N., Bernard, C., Lalonde, R., & Rebaï, M. (2006). The effects of inversion and eye displacements of familiar and unknown faces on early and late-stage ERPs. International Journal of Psychophysiology, 62(1), 141-151.

Eimer, M. (2000b). Event related potentials distinguish processing stages involved in face perception and recognition. Clinical Neurophysiology, 111, 694–705.

Galusca, C. I., Fang, W., Wang, Z., Zhong, M., Sun, Y. H. P., Pascalis, O., & Xiao, N. G. (2022). The “Fat Face” illusion: A robust adaptation for processing pairs of faces. Vision Research, 195, 108015.

Henson, R. N., Goshen-Gottstein, Y., Ganel, T., Otten, L. J., Quayle, A., & Rugg, M.D. (2003).

Electrophysiological and haemodynamic correlates of face perception, recognition and priming. Cerebral Cortex, 13, 793–805.

Hinojosa, J. A., Mercado, F., & Carretié, L. (2015). N170 sensitivity to facial expression: A meta-analysis. Neuroscience & Biobehavioral Reviews, 55, 498-509.

Itier, R. J., & Taylor, M. J. (2004). N170 or N1? Spatiotemporal differences between object and face processing using ERPs. Cerebral cortex, 14(2), 132-142.

Kingdom, F. A., Yoonessi, A., & Gheorghiu, E. (2007). The Leaning Tower illusion: a new illusion of perspective. Perception, 36(3), 475-477.

Piepers, D. W., & Robbins, R. A. (2012). A review and clarification of the terms “holistic,”“configural,” and “relational” in the face perception literature. Frontiers in psychology, 3, 559.

Rossion, B., & Jacques, C. (2011). The N170: Understanding the time course of face perception in the human brain. The Oxford handbook of ERP components, 115-142.

Minor: 

Introduction: 

- Page 3, line 2 – brief explanations for “configural processing” and “featural processing” would make the following points easier to follow. 

 Authors’ response: 

We now define featural and configural processing in the Introduction on Page 3.

- Page 3 – “inversion has a variety of effects on face perception…”, better to specify the sort of effects, e.g. disruption. 

Authors’ response: 

We have added a new section to Page 3 better specifying the disruptive effects of inversion on face processing. 

- Page 3, end of 1st paragraph – did you find reverse illusion for body as well, then why the following sentence claimed this effect tend to be specific for faces? There needs a bit more clarification. 

Authors’ response: 

Thank you. On pages 3-4, we now provide more detailed information on the findings of our previous study (Walsh, Vormberg, Hannaford, & Longo, 2018). We have added a new section to outlining our previous findings on a reverse illusion for bodies. The direction of the size illusion for faces and bodies is opposite – we now make this clearer by adding the phrase “and its directionality” to the following sentence on Page 4: “This illusion and its directionality thus appear to be specific to faces and provides insights into the perceptual and potential neural mechanisms of configural processing.”

Reference

Walsh, E., Vormberg, A., Hannaford, J., & Longo, M. R. (2018). Inversion produces opposite size illusions for faces and bodies. Acta Psychologica, 191, 15-24.

- Page 4, 2nd paragraph – information about the N2pc seems too little, and corresponding interpretation of results from the cited paper was insufficient. Hard to see the necessity of including this reference. 

 Authors’ response: 

We have added information about the N2pc (page 6) – see our response to your first comment above.

Method: 

- For the power analysis, it would be better to specify what are “the t-tests” (e.g. one-way, within-subject, etc”. I just wasn’t sure about what happened here based on the text. 

 Authors’ response: 

We now specify paired t tests on page 6. Also, in response to a related comment from Reviewer 1, we have also performed a sensitivity analysis to determine the smallest effect size our design has the power to detect (also on Pg. 6).

- For each trial, the upright and inverted faces were both in the same expressions, right? Please specify this. 

Authors’ response: 

Yes, this is correct. We have added the following sentence to page 10 to make this clearer: “On each trial, the upright and inverted face identity was the same (though the size could differ) and with the same emotional expression.”

We have also added this information to the caption of the new Figure 2. 

Overall: 

- The paper mentioned “as in our previous study” multiple times, there should be more detailed information regarding what is in the previous paper to help with reading of the current one. If the current paper already provided enough information, then maybe just reduce mentions of the previous paper.

Authors’ response: 

Thank you; we have now added more detail on the findings of our previous paper on Page 4 – see also our above response.

The authors thank Reviewer 2 for their helpful comments which help strengthen our paper.

---

## [Editor Report · Decision Letter 1]

23 Oct 2023

No Influence of Emotional Expression on Size Underestimation of Upright Faces

PONE-D-23-10095R1

Dear Dr. Walsh,

We’re pleased to inform you that your manuscript has been judged scientifically suitable for publication and will be formally accepted for publication once it meets all outstanding technical requirements.

Kind regards,

Shenbing Kuang, Ph.D.

Academic Editor

PLOS ONE
---

## [Editor Report · Acceptance letter]

27 Oct 2023

PONE-D-23-10095R1 

No Influence of Emotional Expression on Size Underestimation of Upright Faces 

Dear Dr. Walsh:

I'm pleased to inform you that your manuscript has been deemed suitable for publication in PLOS ONE. Congratulations! Your manuscript is now with our production department. 

Kind regards, 

on behalf of

Associate Professor Shenbing Kuang 

Academic Editor

PLOS ONE